# Different phylotypes of *Cutibacterium acnes* cause different modic changes in intervertebral disc degeneration

Weibin Lan[1◉], Xiaomeng Wang[1◉], Xuezhao Tu[1], Xiunian Hu[2], Haichuan Lu[1]*

**1** Department of Spinal Surgery, Affiliated Longyan First Hospital of Fujian Medical University, Longyan City, Fujian Province, China, **2** Department of Radiology, Affiliated Longyan First Hospital of Fujian Medical University, Longyan City, Fujian Province, China

◉ These authors contributed equally to this work.
* luhaichuan8934@163.com

**Data Availability Statement:** All relevant data are within the article and its Supporting Information files.

**Funding:** This work was supported by Fujian Province Natural Science Foundation

## Abstract

### Background

The contribution of Cutibacterium *acnes* (*C. acnes*) infection to intervertebral disc degeneration (IDD) and the antibiotic therapy has evoked several controversies in recent years. While some microbiology studies report bacterial disc infection within IDD patients, others attribute the positive results to contamination during prolonged cultures. In addition to the clinical controversy, little was known about the mechanism of *C. acnes*-caused Modic changes (MCs) if *C. acnes* was the pathogenic factor.

### Objectives

This study aimed to investigate the inflammatory mechanism of MCs induced by different phylotypes of *C. acnes* in patients with IDD.

### Methods

Specimens from sixty patients undergoing microdiscectomy for disc herniation were included, *C. acnes* were identified by anaerobic culture, followed by biochemical and PCR-based methods. The identified species of *C. acnes* were respectively inoculated into the intervertebral discs of rabbits. MRI and histological change were observed. Additionally, we detected MMP expression in the rabbit model using reverse transcription-quantitative polymerase chain reaction (RT-qPCR).

### Results

Of the 60 cases, 18 (30%) specimens were positive for *C. acnes*, and we identified 4 of 6 defined phylogroups: IA, IB, II and III. The rabbits that received Type IB or II strains of *C. acnes* showed significantly decreased T1WI and higher T2WI at eighth weeks, while strain III *C. acnes* resulted in hypointense signals on both T1WI and T2WI. Histological examination results showed that all of the three types of *C. acnes* could cause disc degeneration and endplates rupture. Moreover, endplate degeneration induced by type IB or II strains of *C.*

(2019J01051964) and Startup Fund for scientific research, Fujian Medical University(2018QH1238). The funders had no role in study design, data collection and analysis, decision to publish, or preparation of the manuscript.

**Competing interests:** The authors have declared that no competing interests exist.

acnes is related with MMP13 expression. Meanwhile, strain III *C. acnes* might upregulated the level of MMP3.

## Conclusion

This study suggested that *C. acnes* is widespread in herniated disc tissues. Different types of *C. acnes* could induce different MCs by increasing MMP expression.

## Introduction

Intervertebral disc degeneration (IDD) and the subsequent appearance of Modic changes (MCs) in or near the vertebral endplates are usually associated with low back pain (LBP). Epidemiological studies suggested that latent infection of the low-virulent anaerobic bacteria, especially *Cutibacterium acnes* (*C. acnes*), a common skin commensal which was formerly called *Propionibacterium acnes* (*P. acnes*), was a possible cause of IDD, MCs, sciatica and low back pain [1–3]. This raised interesting therapeutic propositions, and supported the use of antibiotics in the treatment of LBP. Previous studies have demonstrated that the prevalence of *C. acnes* in intervertebral disc ranged from 13% to 44%, which is supported by the isolation of this bacterial species from surgical specimens of herniated discs [3–5]. Moreover, the histological identification of *C. acnes* in the intervertebral discs from patients with IDD has provided evidence of its infection, rather than microbiologic contamination [5, 6].

According to other researchers' reports, *C. acnes* is composed of a number of distinct evolutionary lineages, designated types IA, IB, II, and III, based on single locus phylotyping [7, 8]. Based on new technologies such as Multilocus Sequence Typing (MLST) and whole genome sequencing [9], this classification has been extended to IA1, IA2, IB, IC, II, and III types. Furthermore, several previous studies have suggested that these phylogroups may have different pathogenicity traits and may be related to various clinical disease states [10–12]. Nevertheless, little was known about the differences of pathological roles of each subtype of *C. acnes* infected in the intervertebral discs.

Early cartilage endplate degeneration could induce extracellular matrix (ECM) degradation and release of matrix metalloproteinases (MMPs). Thus changes in MMPs are important features of endplate degeneration. It has been reported MMP-13 and MMP1 are associated with MCs and endplate degeneration which caused by *C.acnes* infection in animal models [13, 14]. However, the relationship between different types of *C.acnes* and MMPs remains to be further investigated.

The objectives of the present study were to characterize the prevalence of phylogroups of *C. acnes* in resected disc tissues of patients with lumbar disc herniation, and to compare the pathological changes when inoculated different types of *C. acnes* into the discs of New Zealand white rabbits.

## Materials and methods

### Study subjects and sample collection

A total of 60 patients who underwent lumbar microdiscectomy between January 2019 and November 2019 at The First Hospital of Longyan (Fujian Province, China) were included and consented for participation in this study. All the patients were diagnosed with lumbar disc herniation by MRI. Patients were excluded if they had been received antibiotic treatment within

two weeks of this study start date. Other exclusion criteria were: immunocompromised; traumatic hernia; inflammatory or rheumatic diseases; the presence of unexplained radiological masses. Among included patients, the average age was 54.30 ± 13.48 years, and 33 patients were male and 27 patients were female. All disc samples were obtained using standard operating practices. The specimens were placed aseptically in separate closed sterile glass vials to minimize the possibility of contamination after resection. The size of the excised disc specimen was 3x4x5 mm ~ 10x4x5 mm. This study was approved by the Clinical Ethics Committee of Longyan First Hospital.

## Microbial culture

The tissue samples were cut into smaller pieces, and the tissue broken apart and ground up, using an separately packaged sterile scalpel. The samples were first spread on the surface of a Columbia blood agar (CBA) plate (Oxoid, UK), and then collectively embedded in the center of the plate. For each individual tissue sample, one piece of tissue was used for aerobic incubation and the other piece is used for anaerobic culture. The Anaerobic Work Station Concept 400 (Ruskinn Technology) was used for culture; inoculated plates were cultured for 14 days at 37˚C, 80% N2, 10% CO2, and 10% H2. The same amount of homogenate was also cultured on Columbia Blood Agar (Oxoid) for seven days at 37˚C to detect aerobic bacteria.

## Molecular and phenotypic identification

The colonies were subcultured on CBA plates and cultured under aerobic or anaerobic conditions at 37˚C for 24h, then Gram's staining was performed. The presumptive *C. acnes* isolates were identified using the Rapid ID 32A kit (bioMerieux, France). *Staphylococcus* spp. were identified by standard biochemical tests, latex agglutination for clumping factor/protein A was used to distinguish S. aureus from coagulase negative staphylococci. After purification of genomic DNA using the QIAamp DNA mini kit (Qiagen), *C. acnes* was confirmed by 16S rRNA-based PCR using the primers and conditions previously described [15].

## *recA* sequence analysis and mAb typing

*C. acnes* could be divided into phylogroups IA, IA/IB, IC, II, or III according to Nucleotide sequence analysis of the *recA* housekeeping gene [7, 8]. The *recA* locus was amplified with the previously described primers PAR-1 and PAR-2 for the downstream and upstream flanking sequences of the recA open reading frame, respectively, and the 1201 bp amplicon was generated. According to the manufacturer's instructions, the sequencing process was carried out using the ABI PRISM ready reaction terminator cycle sequencing kits (version 1.1; Perkin-Elmer Applied Biosystems). The samples were analysed on capillary electrophoresis system of ABI PRISM 3100 genetic analyzer (Perkin-Elmer Applied Biosystems). Monoclonal antibody (mAb) typing was performed by immunofluorescence microscopy (IFM) as previously described [7]. *C. acnes* isolates were detected with the mouse mAbs QUBPa1 and QUBPa2. Slides were observed by Leitz Dialux 20 fluorescence microscope.

## Animals

A total of nine New Zealand White rabbits, with an age of about 3 months and an average weight of 2 kg, were used in this study. Spinal deformity was excluded by X-ray examination. Rabbits were randomly divided into three groups with 3 animals in each group (Table 1). The study was approved by the institutional review committee and ethics committee of the authors'

**Table 1. Animal group information.**

| | Inoculated bacterial type | Numbers | Inoculated segment | Bacterial load |
|---|---|---|---|---|
| Group I | The type IB strain of *C. acnes* isolated from patient | 3 | L5-L6 | *C. acnes* suspension in sterilized TSB without bovine serum at $1\times10^7$ CFU/mL with 25μL |
| | | | L4-L5 | Sterilized TSB without bovine serum with 25μL |
| Group II | The type II strain of *C. acnes* isolated from patient | 3 | L5-L6 | *C. acnes* suspension in sterilized TSB without bovine serum at $1\times10^7$ CFU/mL with 25μL |
| | | | L4-L5 | Sterilized TSB without bovine serum with 25μL |
| Group III | The type III strain of *C. acnes* isolated from patient | 3 | L5-L6 | *C. acnes* suspension in sterilized TSB without bovine serum at $1\times10^7$ CFU/mL with 25μL |
| | | | L4-L5 | Sterilized TSB without bovine serum with 25μL |

institution. Rabbits were fed with regular water and food and housed for at least one week before experimentation.

## Preparation of bacterial inoculum

We identified three strains of *C. acnes* isolated from patients with IDD. *C. acnes* harvested from each colony on the agar were washed using TSB without bovine serum for 3 times, then the concentration was adjusted to $1\times 10^7$ CFU (Colony Forming Units) per mL by using the plate count method.

## Percutaneous puncture

All rabbits were anesthetized using pentobarbital sodium for 15 mg/kg via auricular vein and were placed in right lateral position. All operations were performed under sterile conditions. Under the fluoroscopic guidance, percutaneous puncture was performed, starting from the right side of animal, into subchondral bone superior to the L4-L5 and L5-L6 discs. A 22-gauge needle was used with a so-called screw-in method to penetrate cortical bone. The needle tip entered the subchondral bone at a distance of 1 to 2 mm from the adjacent endplate. When the needle tip arrived at the correct location, 25 μL *C. acnes* suspension was inoculated slowly into the nucleus pulposus at the L5-L6 segment and 25 μL of bovine serum-free sterilized TSB was injected into the L4-L5 segment as internal control (Table 1). After the procedure, rabbits were housed individually with free access to food and water and no antibiotics were used before and after the operation.

## MRI examination

Lumbar MRI scans were performed before and every two weeks after the surgery until the eighth week. Animals were fasted on a platform after deep anaesthesia with pentobarbital sodium and scanned with knee coil using the 3.0-T GE HDxt Signa/MRI system (GE Corporation, Connecticut, USA). The scanning parameters: T1W TE/TR, 8.4 ms/540 ms, T2W TE/TR, 48.2 ms/2000 ms, slice thickness, 3 mm, and Field of View, 20 × 20. The MRI scans were evaluated by an experienced radiologist and a spinal surgeon. Besides, we also performed quantitative measurements of the end-plate region signal intensity. ImageJ software (version 1.51; National Institutes of Health), was used to measure signal intensity of the superior subchondral bone region adjacent to L4-L5, and L5-L6 on T2 and T1-weighted central images, as previously reported [16].

## Histological examination

At the eighth week, the animals were euthanized with an overdose of pentobarbital sodium (40 mg/kg) for histological examination. After the intervertebral discs with endplates were fixed in 4% formaldehyde for 24 hours, it was decalcified with Ethylenediaminetetraacetic Acid for three months, processed with routine paraffin embedding, and sectioned at 5 μm. Hematoxylin and eosin (HE) staining was conducted and photographed under the magnification of 100x.

## Cocultures of cartilaginous endplate and *C. acnes*

Following the sacrifice of 3-months-old rabbits under pentobarbital sodium anesthesia, the cartilaginous endplate tissues of rabbits were harvested and cultured. The tissues were carefully excised from intervertebral discs and rinsed in PBS, then were cut into pieces quickly. Endplate cells were obtained by digestion with type II collagenase for two hours at 37˚C. The supernatant was centrifuged at 900 rpm for 5 min and the cellular pellet was then resuspended by using Dulbecco's modified Eagle's medium (DMEM) with 10% fetal bovine serum (FBS). The cells were cultured in condition of 37˚C, 5%$CO_2$. Three strains of *C. acnes* were cultured independently in broth for 2 weeks, the supernatant was harvested by centrifugation at 6000 rpm for 10 min, filtered by using a 0.22-μm filter and then stored at 4˚C. The supernatant of *C. acnes* was added to the endplate cell culture in a 6-well plate.

## RT-qPCR

The endplate cells were cultured with three phylotypes of *C. acnes* respectively for 24h. They were lysed and total RNA was extracted by using Ultrapure RNA kit. Reverse transcription was then performed using HiFiScript cDNA Synthesis kit (CW2569, CWBIO, China). The qPCR was performed to evaluate genes of MMP13 and MMP1, using the UltraSYBR Mixture (CW0957, CWBIO, China). The following primers were included: MMP13 (Forward: AAAG GAGGAGGTTCCTAGAGG; Reverse: ATACCCAATAAAATGTTGGAT), MMP1(Forward: TAGCT GGTTCAACTGCAGGAA; Reverse: AGGCTCACAGAATACATTGGG). Amplification was performed at 95˚C for 10 min (pre-incubation), 40 cycles of 95˚C for 15s, 60˚C for 60s, 72˚C for 20 s (amplification), 95˚C for 15s and 60˚C for 60s (melting curves), and 4˚C for 5 min (cooling).

## Statistical analysis

Analysis of variance (ANOVA) or Student's t-test using SPSS 19.0 was performed to compare mean data among groups, the differences between each group were analyzed with the least significant difference. Statistical significance was taken as $P < 0.05$.

# Results

## Bacterial culture analysis

The results of culture of surgically resected disc specimens showed that bacterial growth was observed for 21 of 60 patients (35%), the other 39 cases had no colonies formation. Of the 60 samples, 18 (30%) specimens were positive for *C. acnes*, and colony numbers of each sample ranged from 1–100 CFU. Coagulase-negative *Staphylococcus* spp was present in 3 cases (5%). For specimens with growth of *C. acnes*, a majority (16/18, 88.9%) yielded only this organism, and there were no additional bacterial species present. Only two patients had tissue samples containing both *C. acnes* and coagulase-negative staphylococci. Aerobic culture did not lead to any additional significant findings.

### Phylotyping of *C. acnes* isolates from intervertebral discs

By using *recA* sequence analysis, it was possible to accurately divide *C. acnes* isolates into types IA (IA1 and IA2), IB, IC, II, or III. Of the 18 strains of *C. acnes*, 39% were found to belong to the type II lineage, identified in 7 cases, 33% were type IA1, present in 6 cases, 22% type III in 4 cases, and 6% type IB in 1 case. No isolates were found to belong to the type IC lineage.

### Signal changes in MRI at intervertebral discs in three groups of rabbits

It has been reported that inoculation of caudal discs with strain IA of *C. acnes* increased intervertebral disc degeneration in rat models [17]. Thus we dismissed this strain of *C. acnes* in this paper, the other three phylotypes of *C. acnes* suspension: type IB, type II and type III, were inoculated into three groups of rabbits, as shown in Table 1. No abnormal signals were found before the operation. Nevertheless, intervertebral discs injected with *C. acnes* showed significant signal changes since the second week in all three groups (Fig 1). Interestingly, hypointense signals at T1WI and hyperintense signals at T2WI in the cartilage endplates area appeared in Group I and Group II, and there were no significant differences in the signal changes between these two groups. While signals on both T1WI and T2WI were hypointense compared to TSB-injected discs in the rabbits of Group I. Furthermore, on T2WI, the normal signal of nucleus pulposus shifted into hypointense signal, indicating the disc degeneration for all three groups. All of these abnormal signals remained constant until the euthanasia of the rabbits at the eighth week, and the volume of signal changes had increased from the second week to the eighth week (Table 2). In contrast, the TSB inoculated control discs were normal during the

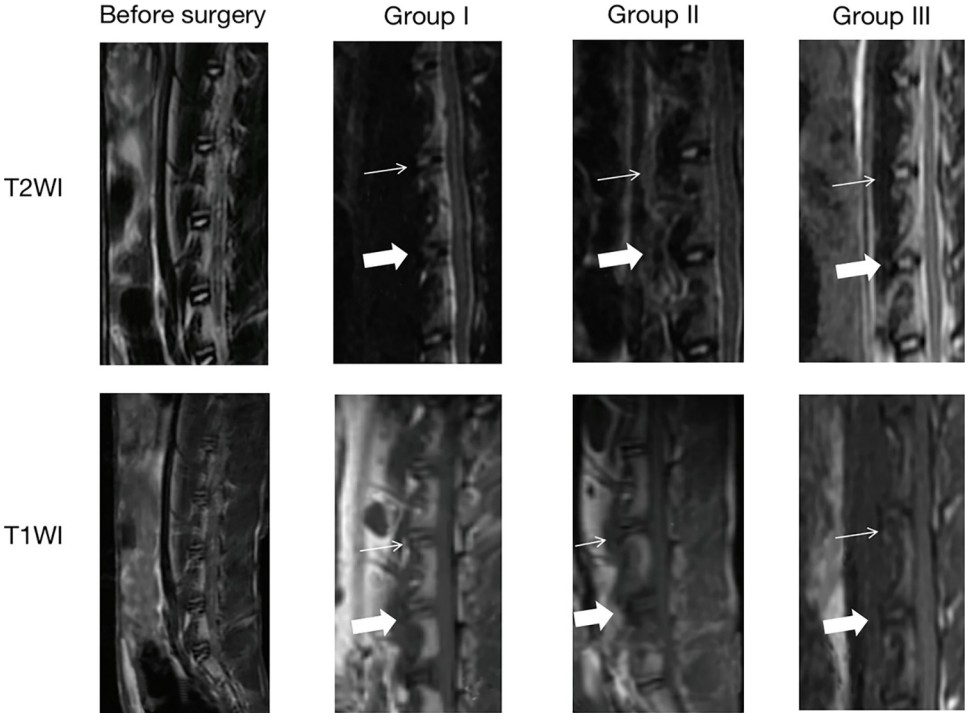

**Fig 1. Signal changes were observed at the second week after the inoculation of the isolated three types strain of *C. acnes*.** Before the surgery, there were no abnormal signals at T1WI and T2WI. An obvious hypointense signal was observed on T1WI and the hyperintense signal was found on T2WI at the *C. acnes*-inoculated segment (L5-L6, indicated by a white filled arrow) in Group I and Group II. The hypointense signal changes were also observed on T1WI in the rabbits of Group III, while the signal intensity of T2WI decreased in this group animals. No significant signal changes were observed at the internal control segment of L4-L5 in all groups (indicated by a thin arrow).

**Table 2. Signal intensity (adjusted by cerebrospinal fluid) of endplate region.**

| | Before injection | After injection* | | | P value |
|---|---|---|---|---|---|
| | | 2w | 4w | 8w | |
| T1WI | | | | | |
| TSB | 2.61±0.48 | 2.67±0.29 | 2.28±0.14 | 2.54±0.68 | 0.306 |
| Group I | 2.72±0.37 | 1.60±0.72 | 1.21±0.85 | 1.03±0.37 | 0.034 |
| Group II | 2.68±0.23 | 1.75±0.64 | 1.34±0.57 | 0.82±0.37 | 0.028 |
| Group III | 2.83±0.18 | 1.52±0.48 | 1.13±0.22 | 0.76±0.45 | 0.019 |
| T2WI | | | | | |
| TSB | 0.41±0.10 | 0.48±0.06 | 0.54±0.07 | 0.68±0.09 | 0.487 |
| Group I | 0.39±0.09 | 0.43±0.04 | 0.52±0.06 | 0.76±0.10 | 0.039 |
| Group II | 0.42±0.10 | 0.48±0.07 | 0.54±0.08 | 0.79±0.04 | 0.025 |
| Group III | 0.43±0.08 | 0.41±0.09 | 0.38±0.10 | 0.31±0.05 | 0.043 |

*The values are given as the mean and the standard deviation.

observation. According to the Pfirrmann classification system, the severity of the intervertebral disc degeneration in the three groups of animals was V or IV grades.

## Intervertebral disc degeneration and endplates rupture after inoculation of *C. acnes*

In contrast with the control group, the segments injected with *C. acnes* showed moderate degeneration with the disappearance of nucleus pulposus and disorganizing of annulus fibrosus at the eighth week in all three rabbits of Group II, while Group I and Group III animals presented disorganized collagen fibers in discs infected with *C. acnes*. The area occupied by the nucleus pulposus was decreased and had an irregular shape. There was endplate rupture at *C. acnes* treated segments in two of Group I animals, one of Group III rabbits and all of three rabbits in Group II. By contrast, TSB injected intervertebral discs had a normal structure, in which nucleus pulposus was enclosed with normal annulus fibrosus (Fig 2). Histological analysis suggested that there was no indication of inflammatory cells in three groups of animals.

## MMP expression in endplate chondrocytes induced by different phylotypes of *C. acnes*

To further investigate the mechanism of Modic changes induced by *C. acnes*, we tested the relationship between MMP and *C. acnes* by stimulating endplate chondrocytes with *C. acnes*

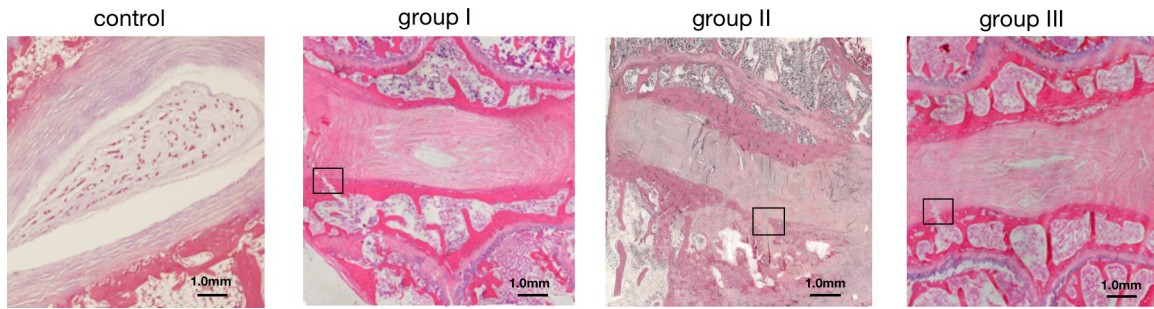

**Fig 2. Typical histological images of slices from the three groups of rabbits at eighth weeks after the inoculation.** The intervertebral discs from the TSB-inoculated control segment (L4-L5) had distinct nucleus pulposus and normal arranged annulus fibrosus. The segment of three types of *C. acnes* inoculated intervertebral discs (L5-L6) was demonstrated as disappearance of nucleus pulposus, endplates fracture (black rectangle), disorganized annulus fibrosus, and partly cartilage proliferation.

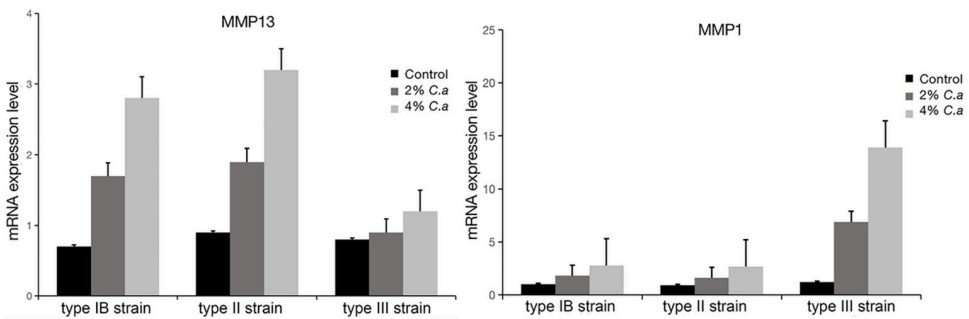

**Fig 3. RT-qPCR analysis of MMP13 and MMP1 mRNA expression in endplate chondrocytes cocultured with 2% or 4% supernatant of different phylotypes of *C.acnes*.** $^*p < 0.05$, $^{**}p < 0.01$.

supernatant. The results showed that the gene expression of MMP13 was increased in type IB and II strains of *C. acnes* supernatant group, while the level of MMP1 in endplate chondrocytes was significantly increased in type III strain of *C. acnes* group compared with controls. All the increase tended to be greater when the concentrations of *C. acnes* was higher (Fig 3). Interestingly, type IB and II strains of *C. acnes* did not effect the expression level of MMP1 in endplate cells, and coculturing with type III strain of *C. acnes* supernatant seems enhance the gene expression of MMP13 but not significantly.

## Discussion

As a member of normal skin and oral microbiota, *C. acnes* may be present in a low number in the blood as a consequence of tooth brushing and endodontic therapy. Since the avascular nuclear disc might provide an ideal anaerobic environment for infection of *C. acnes* [18], it has been proposed that *C. acnes* may cause chronic and mild infection in herniated intervertebral discs. In the present study, *C. acnes* was shown to colonize non-pyogenic disc, with a prevalence rate of 30%, consistent with the previously reported a prevalence of 32% [5, 3, 19]. Some researchers believed that *C. acnes* isolated from intervertebral discs may reflect intraoperative contamination or be pollutants during bacterial culture [20]. However, we believed that most of the bacteria isolated from discs were likely to represent the original growth due to the following reasons. First of all, more than ten research groups have independently proved the presence of *C. acnes* in microbial culture by molecular analysis and histological examination [1, 5, 21]. Therefore, it seems unreasonable to attribute all isolated *C. acnes* as pollutants. Secondly, two histological reports [5, 6] revealed that *C. acnes* grew in clusters within the tissues rather than on the surface of tissues, indicating the growth of bacterium inside intervertebral discs. Overall, the latent residence of *C. acnes* in intervertebral discs should not be ignored, and its role in the pathophysiology of IDD and herniation requires more comprehensive analysis.

The main pathological function of latent infection of *C. acnes* was the induction of IDD. It has been proved that inoculation with *C. acnes* could initiate or accelerate IDD in animal models [22, 23]. However, little is known about the relationship between different phylotypes of *C. acnes* and modic change. Although Modic changes are thought to be associated with disc degeneration, not all Modic changes are found clinically together with degeneration. Interestingly, a previous study showed that the typical strain of *C. acnes* (ATCC 6919) caused Modic type-II changes in rabbit models [24]. It is possible that the early effects of *C. acnes* are so weak for MRI to fully reflect, because a previous research has suggested that MRI may miss Modic change-like pathology [25]. The results of our study indicated that infection of different

phylotypes of *C. acnes* might be capable of producing discal damage differently. We speculated one of the possible reasons may be the different virulence of various species of *C. acnes*, some of them are more virulent, but others may be less virulent [26], thus probably leading to different pathological change and dispersion of MRI image. Therefore, we supported that the intervertebral discs infected by *C. acnes* would have more chance to result in latent low-grade chronic inflammation [3], which is then represented as Modic changes and disc degeneration rather than the pyogenic discitis.

Besides, some limitations still exist in this experiment. The quadrupedal rabbit model might not completely represent biomechanical effects found in humans. The etiology of different types of Modic changes may be continuous. Further studies of primates over a long-term period and more intervening factors would help to deepen our understanding of the occurrence and development of Modic changes. In addition, the sample sizes of both clinical research and animal experiment were relatively small. We agree that even after stringent preoperative skin disinfection, contamination of biopsy samples with skin microbiota cannot be fully excluded. In the part of clinical sample collection, some muscle and ligaments around the disc should be obtained as control tissue in consideration of rigorousness. Further comprehensive investigations about this issue both in vivo and in vitro should be conducted in the future.

In general, we demonstrated a high prevalence of *C. acnes* in intervertebral disc tissue of patients undergoing microdiscectomy. Moreover, according to Koch's Postulates, *C. acnes* might be a potential cause of disc degeneration and Modic changes when inoculated into intervertebral discs. Different strains of *C. acnes* could probably lead to different dispersion of MRI image. In clinical practice, the difference between different types of modic changes still needs to be further investigated.

## Supporting information

**S1 File.**
(XLSX)

## Author Contributions

**Data curation:** Weibin Lan.

**Formal analysis:** Weibin Lan, Xiaomeng Wang.

**Funding acquisition:** Haichuan Lu.

**Investigation:** Xuezhao Tu.

**Methodology:** Xiaomeng Wang.

**Supervision:** Haichuan Lu.

**Visualization:** Xiunian Hu.

**Writing – original draft:** Xiaomeng Wang.

**Writing – review & editing:** Haichuan Lu.

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
