## [Decision Letter · Decision Letter 0]

22 Mar 2022

PONE-D-21-33277Different phylotypes of Propionibacterium acnes cause different modic changes in intervertebral disc degenerationPLOS ONE

Dear Dr. Lu,

Thank you for submitting your manuscript to PLOS ONE. After careful consideration, we feel that it has merit but does not fully meet PLOS ONE’s publication criteria as it currently stands. Therefore, we invite you to submit a revised version of the manuscript that addresses the points raised during the review process.

The authors are called upon to respond to all reviewers' concerns and queries. Specifically, please clarify the categorization of groups and the denomination utilized for the bacterial strain. The MMP related data needs to be further proccessed to incorporate to article. The details regarding experimental proccess need to be clarified at points as well as the utilization of the patient data. Likewise, the disccussion part needs to be edited and specific claims less decisively expressed.

We look forward to receiving your revised manuscript.

Kind regards,

Dragana Nikitovic, Ph.D

Academic Editor

PLOS ONE

Journal Requirements:

2. To comply with PLOS ONE submissions requirements, in your Methods section, please provide additional information on the animal research and ensure you have included details on (1) methods of sacrifice, (2) methods of anesthesia and/or analgesia, and (3) efforts to alleviate suffering.

This work was supported by Fujian Province Natural Science Foundation of China (2019J01051964) and Startup Fund for scientific research, Fujian Medical University.

This work was supported by Province Natural Science Foundation of China (2019J01051964) and Startup Fund for scientific research, Fujian Medical University.The funders had no role in study design, data collection and analysis, decision to publish, or preparation of the manuscript.

Reviewers' comments:

Reviewer's Responses to Questions

**Comments to the Author**

1. Is the manuscript technically sound, and do the data support the conclusions?

Reviewer #1: Yes

Reviewer #2: Yes

2. Has the statistical analysis been performed appropriately and rigorously? 

Reviewer #1: Yes

Reviewer #2: Yes

3. Have the authors made all data underlying the findings in their manuscript fully available?

Reviewer #1: Yes

Reviewer #2: Yes

4. Is the manuscript presented in an intelligible fashion and written in standard English?

Reviewer #1: Yes

Reviewer #2: Yes

5. Review Comments to the Author

Reviewer #1: Dear Authors Manuscript PONE-D-21-33277

Thanks for your submission and the chance to review your work.

It’s an interesting study, using multiple methods: retrieval and culture from surgical intervertebral disc samples from 60 patients undergoing diskectomy; there were P Acnes growth in 18 cases. Different P. Acnes species were identified, and three types were inoculated to three groups of rabbits on the L5-6 disc, having as a control L4-5. Animal spines were imaged q2 weeks via MR and analyzed histologically at 8 weeks. Additionally, discal cells were co-cultured with different types of P Acnes in different concentrations, and MMP expression was measured.

It's an overall well-designed study, interesting and bridging a gap in the field’s knowledge, and have a relevant result (differential effect on the discs of the animal model and culture depending on the P Acnes strain). However, I disagree that different biomechanical environment of the animal model, justifies proposing primates as the next step.

MMP part should be strengthen if it is to be included and reported as results.

I have several observations that may improve the paper:

- P Acnes name has been changed to Cutibacterium acnes, I’d suggest you to at least mention that change either on introduction and/or discussion.

- Animal groups nomination is confusing in the manuscript, sometimes using Groups I, II, III and other times you refer to the bacteria type. Revise.

- How did you measure discal MR signal intensity? Describe your method and the software utilized.

- Page 7, initial paragraph, I guess it should say “Since the avascular…”, not vascular.

- The role of MMP in the pathogenesis of the model is not well introduced, It’s methods for detection are not described and yet, is one of the main results. Please, amend. What’s the hypothetical contribution of MMP to the pathogenesis of MCs?

- Why did you use type IB P Acnes instead of IA1? (actually, you had more cases of IA1). Please, clarify.

- On the part of the discussion, I would moderate the conclusion about type 1-2 P. Acnes producing Modic 1 changes and type 3, Modic 3, since this study has limited numbers of experimental animals for each type. And outside this study, there is no evidence to support such affirmation. I would rephrase this part more on the side of data suggesting that different strains of P Acnes are capable of producing discal damage differently.

- A section of study limitations is missing: low number of subjects (for a prevalence study), not having negative control cultures (i.e. from surgical field), low number of animals per group and thus, using solely the results of the current study, one cannot rule out that P Acnes might be a contaminant. As noted on the discussion as reasons, none of them are contained in this paper. Please add a limitation section on your discussion.

Best,

Reviewer #2: This is well-written well-conducted research in a topic of interest to the spine professionals since colonization by C. acnes in degenerated humans discs is still a question pending an answer. There is both literature supporting this hypothesis and literature going against it. There is even research with molecular analysis of bacterial DNA (NGS) that did not locate C. acnes within the microbiome of degenerated human discs.

In this report, authors isolate C.acnes from human discs after surgical excision and culturing with a 30% positivity rate. Apparently there was no control tissue in this point of the research to determine rate of positive disc colonization by C. acnes, and not contamination during the process. Otherwise, there is controversy as well in controlling culture with surrounding disc tissues (muscles, ligament) since the defect of the annulus could be a way to bacterial spread of the C. acnes outside the disc.

Although this is a weakness to determine rate of true disc colonization, the isolated phylotypes of C. acnes were inoculated into rats intervertebral disc and it does not seem to invalidate this process.

Methods says that patients' pain were accessed but there is no report in the results section about it. If this is not an analyzed measure, it should not be in methods. How long were patients followed after primary surgery?

Percutaneous puncture: it is not clear if after percutaneous puncture of the disc, the rats were opened with a surgical approach to the vertebra and then bone wax was placed at the punctures’ holes. It says that the rat was sutured layer by layer giving the impression that a surgical approach was performed.

This is a good paper considering that is original and brings a new point to the whole C.acnes disc degeneration discussion. It does not prove or gives an answer to the whole C. acnes presence in the disc or its relationship to disc degeneration, but it is original in stating the effects of the C. acnes and some of their different phylotypes in normal disc of rats. Rats spine different biomechanics is a limitation and has been brought into the limitations paragraph.

It is expected that inoculation of any bacteria into a normal disc would lead to infection and inflammatory response and knowing how different types of C. acnes reacts to disc degeneration and its endplates is a good and new data, although the main question on the presence of C. acnes in the human disc remains unanswered.

6. PLOS authors have the option to publish the peer review history of their article (what does this mean?). If published, this will include your full peer review and any attached files.

Reviewer #1: **Yes: **Mauricio Campos-Daziano MD

Reviewer #2: **Yes: **Nelson Astur

---

## [Author Response · Author response to Decision Letter 0]

15 May 2022

Thank you for your letter and the reviwers’ comments concerning our manuscript. We are happy to improve the manuscript based on these helpful and valuable comments from you. We have revised the manuscript according to the editor’s advice, hoping to meet the submission requirements. Revisions in the text are shown using red highlight for additions, and strikethrough font for deletions. The responses to the reviewer's comments are presented in the response file.

---

## [Decision Letter · Decision Letter 1]

22 Jun 2022

Different phylotypes of Cutibacterium acnes cause different modic changes in intervertebral disc degeneration

PONE-D-21-33277R1

Dear Dr. Lu,

We’re pleased to inform you that your manuscript has been judged scientifically suitable for publication and will be formally accepted for publication once it meets all outstanding technical requirements.

Kind regards,

Dragana Nikitovic, Ph.D

Academic Editor

PLOS ONE

Additional Editor Comments (optional):

Reviewers' comments:

Reviewer's Responses to Questions

**Comments to the Author**

1. If the authors have adequately addressed your comments raised in a previous round of review and you feel that this manuscript is now acceptable for publication, you may indicate that here to bypass the “Comments to the Author” section, enter your conflict of interest statement in the “Confidential to Editor” section, and submit your "Accept" recommendation.

Reviewer #1: All comments have been addressed

2. Is the manuscript technically sound, and do the data support the conclusions?

Reviewer #1: Yes

3. Has the statistical analysis been performed appropriately and rigorously? 

Reviewer #1: Yes

4. Have the authors made all data underlying the findings in their manuscript fully available?

Reviewer #1: Yes

5. Is the manuscript presented in an intelligible fashion and written in standard English?

Reviewer #1: Yes

6. Review Comments to the Author

Reviewer #1: Thanks for accepting our suggestions to make your manuscript more publishable. I think it would constitute a good piece for bridging the knowledge gaps remaining the C acnes hypothesis.

7. PLOS authors have the option to publish the peer review history of their article (what does this mean?). If published, this will include your full peer review and any attached files.

Reviewer #1: **Yes: **Mauricio Campos Daziano

---

## [Editor Report · Acceptance letter]

4 Jul 2022

PONE-D-21-33277R1 

Different phylotypes of *Cutibacterium acnes* cause different modic changes in intervertebral disc degeneration 

Dear Dr. Lu:

I'm pleased to inform you that your manuscript has been deemed suitable for publication in PLOS ONE. Congratulations! Your manuscript is now with our production department. 

Kind regards, 

on behalf of

Dr. Dragana Nikitovic 

Academic Editor

PLOS ONE